# Conditionally-Parameterized, Discretization-Aware Neural Networks for Mesh-Based Modeling of Physical Systems

**Jiayang Xu**
davidxu@umich.edu

**Aniruddhe Pradhan**
anipra@umich.edu

**Karthik Duraisamy**
kdur@umich.edu

**Department of Aerospace Engineering, University of Michigan, Ann Arbor, MI 48109.**

## Abstract

Simulations of complex physical systems are typically realized by discretizing partial differential equations (PDEs) on unstructured meshes. While neural networks have recently been explored for the surrogate and reduced order modeling of PDE solutions, they often ignore interactions or hierarchical relations between input features, and process them as concatenated mixtures. We generalize the idea of conditional parameterization – using trainable functions of input parameters to generate the weights of a neural network, and extend them in a flexible way to encode critical information. Inspired by discretized numerical methods, choices of the parameters include physical quantities and mesh topology features. The functional relation between the modeled features and the parameters is built into the network architecture. The method is implemented on different networks and applied to frontier scientific machine learning tasks including the discovery of unmodeled physics, super-resolution of coarse fields, and the simulation of unsteady flows with chemical reactions. The results show that the conditionally-parameterized networks provide superior performance compared to their traditional counterparts. The CP-GNet - an architecture that can be trained on very few data snapshots - is proposed as the first deep learning model capable of standalone prediction of reacting flows on irregular meshes.

## 1   Introduction

Numerical simulations of partial differential equations (PDEs) have become an indispensable tool in the study of complex physical systems. High-resolution simulations are, however, prohibitively expensive or intractable in many practical problems. Machine learning techniques have recently been explored to improve the efficiency and accuracy of traditional numerical methods. Successful applications include nonlinear model order reduction [1, 2, 3], model augmentation [4, 5, 6], and super-resolution [7, 8, 9]. Neural networks have also been used to replace traditional PDE-based solvers, and serve as a standalone prediction tool. Popular approaches include auto-regressive time-series predictions [10, 11, 12, 13, 14], Physics-Informed Neural Networks (PINNs) [15, 16, 17].

Despite promising results on canonical problems, commonly used network architectures such as autoencoders and CNNs have inherent limitations. An autoencoder generates a fixed mapping between the geometric coordinates and the encoded digits. This limits their portability for new geometries and dynamic patterns. A CNN requires interpolation of existing data to a structured, Euclidean space, introducing additional cost and error. Irregular geometry boundaries require constructs such as elliptic coordinate transformation [18] and Signed Distance Function (SDF) [1]. Moreover, models often ignore the hierarchical relations between heterogeneous features, and concatenate them into a single input vector, e.g. the common concatenation of the edge and node features in Graph

Neural Networks (GNNs). The learning of high-order terms remains mostly unguided – even simple quadratic terms are often fitted via a number of hidden units in a brute-force manner.

With a focus on mesh-based modeling of physical systems, we use the idea of conditional parameterization (CP) to build the hierarchical relations between different physical quantities as well as numerical discretization information into the network architectures. The key contributions of our work are as follows [1]:

1. We demonstrate that a drop-in CP modification can bring significant improvements for various existing models on several tasks essential to the modeling of physical systems.

2. We propose a conditionally parameterized graph neural network (CP-GNet), which effectively models complex physics such as chemical source terms, irregular mesh discretizations, and different types of boundary conditions.

3. We conduct extensive numerical tests and demonstrate state-of-the-art performances on problems of different complexities, ranging from the basic viscous Burgers equation to a complex reacting flow.

## 2 Methodology

**Conditional Parametrization:** The idea of conditional parametrization (CP) is to use trainable functions of input parameters to generate the weights of a neural network. To demonstrate this, we start from a standard dense (fully connected) layer:

$$\mathbf{h}(\mathbf{u}; \mathbf{W}, \mathbf{b}) = \sigma(\mathbf{W}\mathbf{u} + \mathbf{b}), \tag{1}$$

where $\mathbf{u} \in \mathbb{R}^{n_x}$ is the input feature vector, $\mathbf{h} \in \mathbb{R}^{n_h}$ is the output hidden state vector, $\mathbf{W} \in \mathbb{R}^{n_h \times n_x}$ and $\mathbf{b} \in \mathbb{R}^{n_h}$ are the trainable weights and bias, and $\sigma$ is the activation function. It can be seen that in the evaluation stage, the values of $\mathbf{W}$ and $\mathbf{b}$ are fixed regardless of the inputs. Thus the performance of Eq. (1) is largely limited by the interpolation range of training data.

By introducing a parameter vector $\mathbf{p} \in \mathbb{R}^{n_p}$ and a trainable function $f(\mathbf{p}) : \mathbb{R}^{n_p} \to \mathbb{R}^{n_h \times n_x}$ that computes the weights $\mathbf{W}$ based on $\mathbf{p}$, the conditionally parameterized version of Eq. (1) is given by:

$$\mathbf{h}(\mathbf{u}; f(\mathbf{p}), \mathbf{b}) = \sigma(f(\mathbf{p})\mathbf{u} + \mathbf{b}). \tag{2}$$

An easy way to incorporate the formulation into existing neural network models is by making $f$ a single-layer MLP, the conditionally parameterized dense (CP-Dense) layer can be represented by:

$$\mathbf{h}(\mathbf{u}, \mathbf{p}; \mathbf{W}, \mathbf{B}, \mathbf{b}) = \sigma\left(\sigma\left(\langle \mathbf{W}, \mathbf{p} \rangle + \mathbf{B}\right)\mathbf{u} + \mathbf{b}\right). \tag{3}$$

It should be noted that this would bring a change in the dimensions of weights and biases, which become $\mathbf{W} \in \mathbb{R}^{(n_h \times n_u) \times n_p}$, $\mathbf{B} \in \mathbb{R}^{n_h \times n_u}$. When the layer width is kept the same, the total number of trainable parameters increases linearly with the parameter size $n_p$. In applications, $\mathbf{p}$ is not limited to an additionally-introduced parameter. When simply taking $\mathbf{u}$ as the parameter for itself, the quadratic terms will be introduced. High-order terms, which are prevalent in physical systems, can be easily modeled using multiple such layers. In Appendix B, we demonstrate how certain discretized PDE terms can be fitted exactly with simple conditionally parameterized layers.

### 2.1 CP-GNet for mesh-based modeling of physical systems

**Graph representation of discretized systems:** Consider a physical system governed by a set PDEs for a time-variant vector of variables $\mathbf{q}(t)$. Using the popular finite volume discretization, the computational domain is divided into contiguous small cells, indexed by $i$. The discretized form of equation can be written as:

$$\frac{d\mathbf{q}_i(t)}{dt} = \frac{1}{\Omega_i} \sum_{j \in N(i)} \mathbf{f}\left(\mathbf{q}_i, \mathbf{q}_j, \mathbf{n}_{ij}\right) A_{ij} + \mathbf{s}(\mathbf{q}_i), \tag{4}$$

---

[1]The source code is released to facilitate future research at `https://github.com/davidxujiayang/cpnets`

where $\mathbf{q}_i$ is the cell-centered value of cell $i$, $\Omega_i$ is the volume (3D)/area (2D) of the cell, and $N(i)$ is the neighborhood set of cells around $i$. Between a neighboring pair of cells $i$ and $j$, $A_{ij}$ is area (3D)/length (2D) of the shared cell boundary, and $\mathbf{n}_{ij} = (\mathbf{x}_i - \mathbf{x}_j)/|\mathbf{x}_i - \mathbf{x}_j|$ is a vector between the cell center locations $\mathbf{x}_i$ and $\mathbf{x}_j$. In the explicit numerical simulation of Eq. (2.1), solutions are updated by computing the increment of $\Delta\mathbf{q}_i^k = \mathbf{q}_i^{k+1} - \mathbf{q}_i^k$ between discrete time steps indexed by $k$, which is determined by two terms. The *flux term* $\mathbf{f}$ computes the exchange of quantity between neighboring cells, which is a complex function involving both the cell values as well as the vector between them, e.g. [19]. The *source term* $\mathbf{s}$ computes physics that are local to the cell, such as the reaction of chemical species.

In our setting, the discretized system is mapped to a graph $G(V, E)$, defined by nodes $V$ of size $|V| = n_v$ connected by edges $E \subset V \times V$ of size $|E| = n_e$. Each node $\mathbf{v}_i$ is located at the corresponding cell center $\mathbf{x}_i$, and each edge $(i, j)$ corresponds to a shared boundary between the finite volume cells. Denoting the sets of mapped quantities on all nodes and edges of $G$, $\mathbf{Q} = \{\mathbf{q}_i, i \in V\}, \mathbf{N} = \{\mathbf{n}_{ij}, (i, j) \in E\}$, the target is to develop a graph neural network operator $g$ that predicts the increment as $\Delta\mathbf{Q}^k = g(\mathbf{Q}^k, \mathbf{N})$.

**CP-GNet architecture:** The architecture for the proposed conditionally parameterized graph neural network, CP-GNet, can be written in an encoder-processor-decoder form. A schematic is provided in Fig. 1. For clarity of different variables in the description of the network, we use $\mathbf{u}_i$ for the latent variables on node $i$ to distinguish from the physical variables $\mathbf{q}_i$, and use $\mathbf{e}_{ij}$ for latent variables on edge $(i, j)$ to distinguish from the vector $\mathbf{n}_{ij}$.

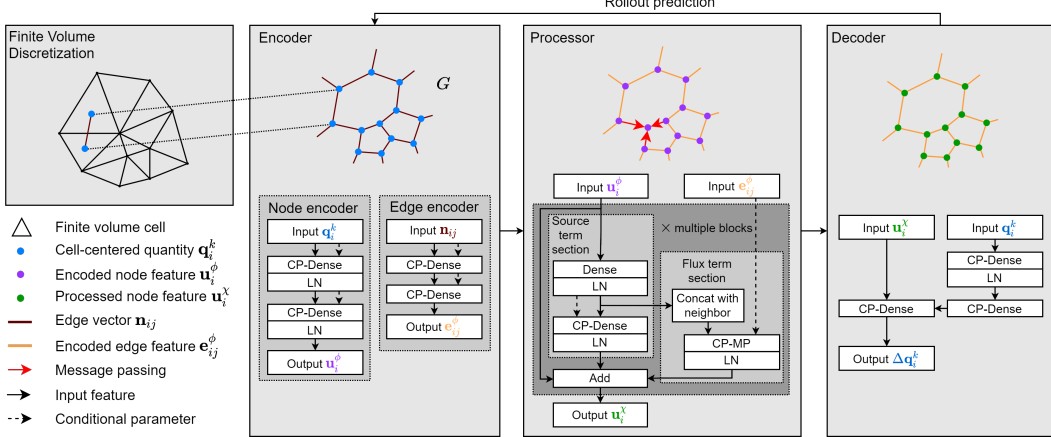

Figure 1: Schematic of CP-GNet architecture

**Encoder:** Numerical solution of PDEs (e.g., the compressible Navier–Stokes equations) requires the processing of arbitrarily complex interactions between mesh elements. While large MLP architectures can represent this complexity, the data requirements to reliably train such networks might be large. In contrast, our proposed encoder takes two CP-Dense layers, taking the output from the previous layer as both the input and the conditional parameter. Through the encoder, high-order interactions can be easily extracted, allowing a degree of extrapolation by virtue of linearity. The CP-GNet uses two separate, but similarly constructed encoders to process the input node features $\mathbf{q}_i^k$ and edge features $\mathbf{n}_{ij}$, respectively.

**Processor:** The flux term $f$ in Eq. (2.1) can be effectively approximated by CP message-passing (CP-MP) between adjacent nodes on a graph. The source term $s$, on the other hand, can be modeled by CP-Dense layers. In CP-GNet the processor consists of multiple identical blocks with independent weights. Residual connections are added between the blocks. As shown in Fig. 1, each block includes a CP-MP based section and a CP-Dense based section to address the two types of terms. Modified from the Edge Conditioned Convolution (ECC) [20], the CP-MP computation is formulated as:

$$\mathbf{W}_{ij} = \sigma\left(\left\langle \mathbf{W}, \mathbf{e}_{ij}^\phi \right\rangle + \mathbf{B}\right), \;\; \mathbf{h}_i = \sum_{j \in N(i)} w_{ij}\sigma\left(\left\langle \mathbf{W}_{ij}, [\mathbf{u}_i; \mathbf{u}_j] \right\rangle\right), \tag{5}$$

where $\mathbf{e}_{ij}^{\phi}$ is the output from the edge encoder, $\mathbf{u}_i$ and $\mathbf{u}_j$ are the latent node features from the previous layer, $\mathbf{h}_i$ is the nodal latent output, and $w_{ij} = A_{ij}/\Omega_i$ is the flux weight from Eq. (2.1).

**Decoder:** It is common in a PDE solver to use a Jacobian matrix $\mathbf{J} = \partial\mathbf{u}/\partial\mathbf{h}$ to transform the variable increments $\Delta\mathbf{u} = \mathbf{J}\Delta\mathbf{h}$. The decoder in the GP-GNet serves a similar purpose – to convert hidden variables to the output on the physical space. Similar to the encoder, the decoder consists of three conditionally parameterized dense layers. The first two layers can actually be viewed as a dedicated "encoder" that is similar to the initial node encoder, taking $\mathbf{q}_i^k$ as the input, but with independent weights. The purpose of this "encoder" is to extract a final conditional parameter, which is used in the third CP-Dense layer in the decoder to determine the weights for the output node feature $\mathbf{u}_i^{\chi}$ from the processor. The third decoder layer is also the final layer of the model, which outputs $\Delta\mathbf{q}_i^k$ (with proper scaling). Except for the edge encoder and the last two layers in the decoder, all dense, CP-Dense, CP-MP layers are appended with LayerNormalization (LN) layers.

**Treatments for boundaries:** The computational domain of a practical problem includes multiple types of boundaries, e.g. the case in Sec. 4.3. In classic PDE solvers, they are treated with different boundary conditions, which define explicit formulations to compute relationships of the domain with the external world. However, these conditions and formulations are only defined for the physical quantities, thus cannot be easily transferred for latent variables, especially when multiple message-passing/convolution steps are used. To enable the GP-GNet to model different types of boundaries efficiently, special treatments are necessary. For boundaries with known inputs, such as the inlet and the outlet, their values are directly input to the corresponding nodes at every time step. For the boundaries imposing certain constraints, instead of a given physical value, such as Neumann and symmetry boundaries, *ghost edges* are introduced. For a cell $i$ with a face lying on a boundary, we introduce a ghost edge vector $\mathbf{n}_{ig}$, that points from the corresponding node $i$ to the center of the boundary face. Ghost edges are processed together with the normal edges in the edge encoder. However, the CP-MP layer in the processor of the CP-GNet is slightly modified. More specifically, the concatenation $[\mathbf{u}_i; \mathbf{u}_j]$ in Eq. (5) is replaced with only $\mathbf{u}_i$. And for each type of boundary, the weights for the CP-MP layer are trained independently, to let the model learn different types of boundary condition for the latent variables. The effectiveness of this treatment is shown in Sec. 4.3 and further discussed in Appendix. A.3.3.

## 3 Related Work

There have been successful attempts towards making networks directly parametric to certain features, such as connectivity patterns [21], layer embedding [22], mean image features [23]. The Conditionally Parameterized Convolution (CondConv) model [23], makes convolution kernel weights as a linear combination of functions of the input features, and achieves an efficient expansion of the network capacity. The Hypernetwork [22] uses a single network that takes layer embeddings, e.g., layer index, to generate the weights for different layers of the main network, and reduced the total number of trainable weights. A popular framework to perform convolution on graphs is the message passing neural network (MPNN) [24], which treats graph convolutions as messages passed between nodes through edges. In this approach, the node features and edge act on intermediate variables, and the output is expressed as a linear combination through concatenation. This can fail when the impact of node features rely on the edge features in a non-linear fashion. To address this, Edge Conditioned Convolution [20] (ECC) makes the weights for node features dependent on edge features. After the modification for conditional parametrization, ECC was shown to achieve excellent performance on irregular point cloud data. In comparison, our method extends the choice of parameters to physical quantities, hidden inputs themselves, as well as discretization information.

Multiple architectures in the family of GNNs have shown successes in processing irregular, non-Euclidean features. Applications include cloud classification [25, 26], action recognition [27] and control [28], traffic forecasting [29, 30], quantum chemistry [24]. Attempts on using GNNs in scientific computation are relatively limited and are mostly focusing on particle-based methods [31, 32]. Recently, pioneering work has demonstrated the potential of using GNNs for mesh-based scientific computation. CFD-GCN [33] coupled a GNN with an existing PDE solver to perform hybrid-fidelity prediction and achieved higher efficiency than traditional high-fidelity solvers. MeshGraphNets [34] extends the encoder-processor-decoder structure from Graph Network-based Simulators (GNS) [32], and demonstrated impressive performance on mesh-based simulations for a wide range of physical systems. Compared to these approaches, our method with CP models the high-order terms and irreg-

ular discretizations more effectively. Appendix C compares our method with the MeshGraphNets on flow simulation tasks.

## 4 Numerical Tests

We applied conditional parametrization to network architectures for three distinct, but important tasks in scientific computing. The first two tasks are on uniform Euclidean grids, and the discretization information is directly included in the conditional parameters such as the differential terms and the local Reynolds number. Comparisons between appropriate baseline models and their CP modifications are performed. The third task uses an irregular mesh with complex boundaries and is conducted with the CP-GNet model we proposed. The non-CP modification, which takes standard dense and message passing layers with more units, is used as the baseline. Appendix A.1 provides more details on the studied system and the generation of data; A.2 provides details on network training; A.3 provides additional results and analysis. An additional test for the flow over a cylinder is performed in the comparison against the MeshGraphNets in Appendix C.

### 4.1 Discovery and solution of coarse-grained models

In many practical problems, high-fidelity simulations are not affordable. Instead, computations are performed using coarse-grained models, e.g. the Large Eddy Simulation [35]. In such models, the small-scale physics are unresolved, and are approximated using additional *closure* terms in the PDEs, the development of which constitutes an important area of research. In fact, even for the seemingly simple (yet richly non-linear) equation presented below, a perfect closure model is unknown. In this work, we demonstrate how CP models can be used to develop a closure model for the coarse-grained 1D viscous Burgers equation that is often used in the study of shock formation, traffic flows, and turbulent interactions, etc. For the unknown spatio-temporal field $u(x, t)$ on a spatially periodic domain $x \in [0, L]$, the original equation is given by:

$$\frac{\partial u}{\partial t} + u \frac{\partial u}{\partial x} - \nu \frac{\partial^2 u}{\partial x^2} = 0, \tag{6}$$

where $\nu$ is a diffusion coefficient and $u(x, 0)$ is a random initial condition (See Appendix A.1.1).

When this equation is solved on a finely discretized mesh, the dynamics can be regarded as fully resolved. However, if a solution is attempted on a coarse mesh with Eq. (6) without any additional treatments, the solution becomes inaccurate and numerically unstable, thus a closure operator $\mathcal{C}(\cdot)$ is needed. Representing the quantity on the lower resolution mesh by $\bar{u}$, the "closed" equation is:

$$\frac{\partial \bar{u}}{\partial t} + \bar{u} \frac{\partial \bar{u}}{\partial x} - \nu \frac{\partial^2 \bar{u}}{\partial x^2} + \mathcal{C} = 0. \tag{7}$$

In this experiment, two baseline models for $\mathcal{C}$ and their CP developments are compared. The first model is 2-layer CNN with a dense layer with ReLU activation, followed by a 1D convolution layer. This model assumes the closure term to be a function of convection term $\bar{u} \frac{\partial \bar{u}}{\partial x}$ and the diffusion term $\nu \frac{\partial^2 \bar{u}}{\partial x^2}$, and takes their concatenation $\mathbf{q} = [\bar{u} \frac{\partial \bar{u}}{\partial x}, \nu \frac{\partial^2 \bar{u}}{\partial x^2}]$ as the input. Its CP variant, CP-CNN, replaces the first layer with a CP-Dense layer that takes $\mathbf{q}$ as the parameter for its own weights. The second baseline model is a reference Data-Driven Parameterization (DDP) model [36]. The model takes $\mathcal{C}$ as a function of the filtered variable $\bar{u}$, which is modeled by an 8-layer MLP with swish activation. Similarly, the CP variant, CP-DDP replaces the first layer with a CP-Dense layer that takes $\mathbf{q}$ as the parameter for the weights for $\bar{u}$. The network architectures are presented in Fig. 2.

Two sets of data are used. The high resolution runs are solved with Eq. (6) from two different initial conditions (ICs) on a shared 2048-grid-node mesh. The low resolution solutions are obtained by applying a box filter to each step of the high resolution solutions onto a 32-grid-node mesh. The ground truth for $\mathcal{C}$ is then computed based on the low resolution data. Each set of data consists of 267 time steps, spanning a period of 2 s. The first 0.2 s of data for one IC is used for training.

Online testing computations are then carried out from the filtered, low resolution ICs using Eq. (7), with $\mathcal{C}$ computed based on the online solution at every time step. $x$-$t$ contours are present in Fig. 3 to compare the evolution of $\bar{u}$. Spatial profiles are also plotted at a few steps to provide more details. Despite a small time step (CFL number$<$ 0.5, without any closure term, the computation

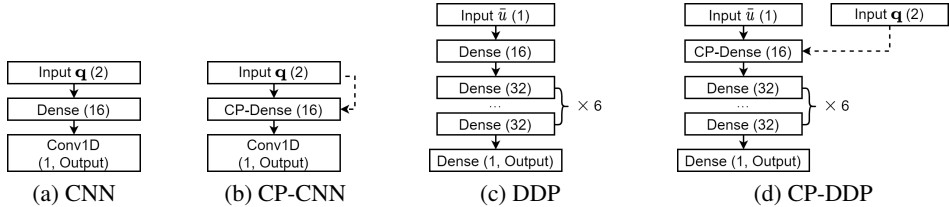

(a) CNN  (b) CP-CNN  (c) DDP  (d) CP-DDP

Figure 2: Closure modeling network architectures. Solid arrow: input feature; dashed arrow: condition parameter; numbers: layer width.

Table 1: Closure model MAE. $\bar{u}$ Avg.: averaged over all steps for online prediction for $\bar{u}$; $\bar{u}$ final: for the final step of online prediction; Inf.: Unbounded cases.

|  | Training IC | | Testing IC | |
|---|---|---|---|---|
|  | $\bar{u}$ Avg. | $\bar{u}$ final | $\bar{u}$ Avg. | $\bar{u}$ final |
| CNN | 0.23 | 0.41 | 0.16 | 0.23 |
| CP-CNN | **0.15** | **0.21** | **0.09** | **0.13** |
| DDP | Inf. | Inf. | Inf. | Inf. |
| CP-DDP | 0.42 | 0.89 | 0.3 | 0.41 |

is numerically unstable and the error grows unbounded. The baseline CNN model is able to keep the solution stable within the period studied, and the CP-CNN improves the accuracy noticeably. The baseline DDP model is only able to postpone the "blow-up" to slightly later. The solution with CP-DDP closure is bounded throughout the period. The improvements are also valid for both the unseen IC. The Mean Absolute Error (MAE) for $\bar{u}$ is provided in Table. 1.

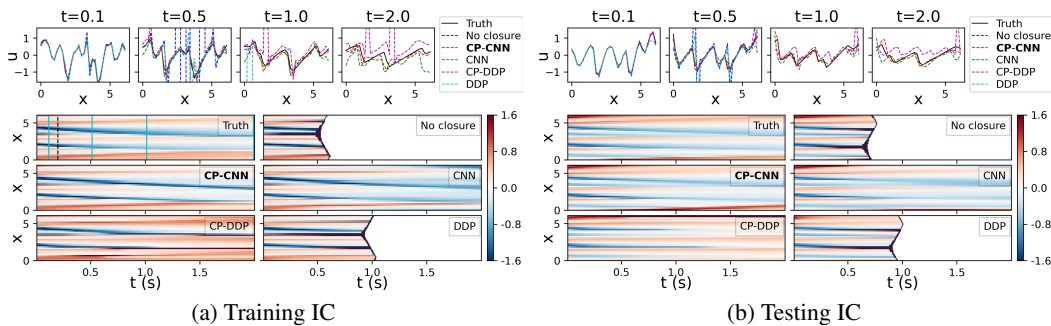

(a) Training IC  (b) Testing IC

Figure 3: Closure modeling results. The first $t \leq 0.2$ s for the left case is used for training, marked by the black dashed line in the first contour. The $x$-$t$ contours show the evolution of $\bar{u}$. **The reference DDP model solution grows into infinity, shown as white areas in the contour**. The gaps between models are more visible in the spatial profiles at time steps marked by the cyan lines.

## 4.2 Super-resolution of chaotic flows

In this experiment, we perform enrichment of low-resolution snapshots of turbulent flow fields. In an enrichment/super-resolution process, one inputs a low-resolution snapshot of the solution, and seeks a snapshot with better resolution. One way to achieve different resolutions on a given mesh is to use Discontinuous Galerkin (DG) projection [37]. In this method, the solution within a mesh element $i$ is represented by coefficients $\mathbf{a}_i$ for a set of polynomial bases, of which the size is determined by the polynomial order $P$. The final resolution of the solution is jointly determined by $P$ and the element width $L$. More specifically, wall-parallel snapshots from the solution of a turbulent channel flow [38] is studied, and the task is to recover high-order ($P = 3$) DG coefficients $\mathbf{a}_i^h \in \mathbb{R}^9$ for the $x$-velocity from lower-order ($P = 1$) ones $\mathbf{a}_i^l \in \mathbb{R}^4$. 5 snapshots are generated in total at different normalized wall-normal heights $z^+ \in \{650, 700, 750, 800, 850\}$, as illustrated in Fig. 4. Each snapshot spans an area of $X \times Y = 2\pi \times \pi$, and is projected onto a shared set of uniform meshes with 6 different widths $L \in \{\pi/4, \pi/8, \pi/12, \pi/16, \pi/24, \pi/32\}$, for the two studied polynomial orders $P \in \{1, 3\}$. Thus,

Table 2: Average and maximum absolute errors in the integral of super-resolved energy spectra.

| | Training | | | | Testing | | | |
|---|---|---|---|---|---|---|---|---|
| | $E_x$ Avg. | $E_x$ Max. | $E_y$ Avg. | $E_y$ Max. | $E_x$ Avg. | $E_x$ Max. | $E_y$ Avg. | $E_y$ Max. |
| MLP | 0.0145 | 0.0391 | 0.0272 | 0.0609 | 0.0098 | 0.0364 | 0.0184 | 0.0675 |
| CP-MLP | **0.0120** | **0.0328** | **0.0217** | **0.0429** | **0.0081** | **0.0260** | **0.0158** | **0.0519** |

for each $z^+$, 12 sets of data, each for one combination of $L$ and $P$, are provided. Fig. 4 shows a few example contours at different combinations for $z^+ = 800$. The data for $z^+ \in 700, 800$ is used for training. It should be noted that the coefficients are computed independently for each mesh element, thus the total number of training points is a few thousand, instead of 24 (which should be multiplied by the number of elements). More details on the data generation are provided in Appendix A.1.2.

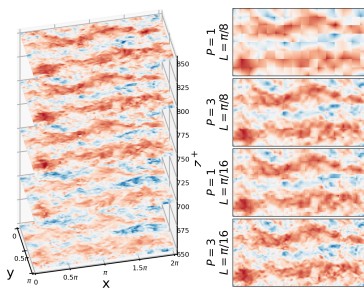

Figure 4: Snapshots for super-resolution

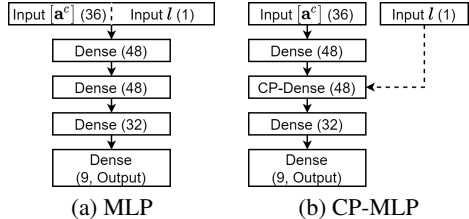

(a) MLP        (b) CP-MLP

Figure 5: Super-resolution network architectures Solid arrow: input feature; dashed arrow: condition parameter; numbers: layer width.

In this task, the baseline model is from the compact super-resolution model by Pradhan and Duraisamy [8]. It takes $\mathbf{a}_i^h$ as a function of two inputs. The first input is a concatenation of normalized low-order basis coefficients for $i$ and its neighbors $N(i)$:

$$[\mathbf{a}^c]_i = [\{\mathbf{a}_j^c - \bar{\mathbf{a}}^c; j \in N(i) \cup i\}]/u_i^{\text{RMS}}, \tag{8}$$

where $[\{\cdot\}]$ denotes the concatenation of all elements in a set, and $\bar{\mathbf{a}}^c$ is the mean of the set. In our case, we include all immediate neighbors, including corner ones in $N(i)$, thus $[\mathbf{a}^c]_i \in \mathbb{R}^{36}$. The second input to the model is an indicator $l_i = \log(Re_i^L)$ for the loss of information in the low-order projection process. $Re_i^L = \frac{u_i^{\text{RMS}}L}{\nu}$ is the local Reynolds number. The indicator reflects that the loss is a function of the kinetic energy, measured by $u_i^{\text{RMS}}$, mesh resolution $L$, and fluid viscosity $\nu$. Because $Re_i^L$ can vary by orders of magnitude across elements, log scaling is used. The two inputs are first concatenated and then processed in a 4-layer MLP in the baseline model. In contrast, the conditionally parameterized model CP-MLP processes only the first input $[\mathbf{a}^c]_i$ in the dense layers. The second dense layer is replaced by a CP-Dense layer, where the second input $l_i$ is instead taken as a conditional parameter for the weights for the latent output of the first layer. A comparison of the model architectures is provided in Fig. 5.

Results for two sample testing cases, $(z^+ = 650, L = \pi/4)$ and $(z^+ = 750, L = \pi/8)$ are shown in Fig. 6. It can be observed that the CP-MLP is able to reconstruct more small scale structures compared with the MLP. The performance can be qualified by the stream-wise and span-wise energy spectra, $e_x$ and $e_y$ (see Appendix A.3.2 for definitions). $e_x$ for different stream-wise wave numbers $k_x$ is shown in Fig. 6. It can be observed that for high-order projection or super-resolution, the high-wave-number spectra are much richer than those for the low-order projection. The CP-MLP plots follow the truth noticeably better than the MLP baseline, which confirms our observation from the contours. Absolute error in the integrals of energy spectra, $E_x = \int_{k_x} e_x dk_x$ and $E_y = \int_{k_y} e_y dk_y$ are computed for the 24 training and 36 testing sets and summarized in Table 2. Both training and testing errors are reduced significantly when CP is applied.

### 4.3 Simulation of reacting flows in a rocket engine injector

We use a highly complex public dataset [39] as a model of combustion processes in a rocket engine injector [40]. The dataset includes solutions on a 2D finite-volume mesh with 308184

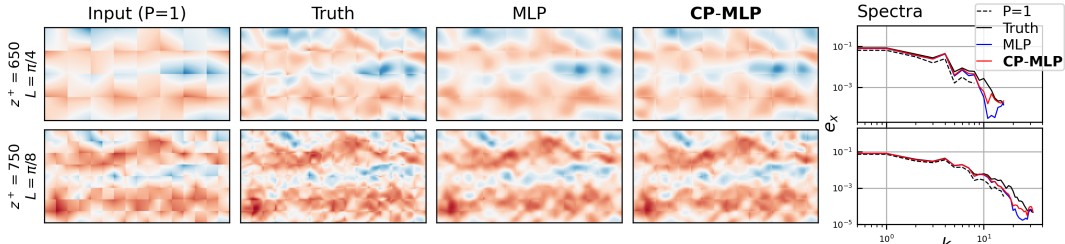

Figure 6: Super-resolved flow field and stream-wise energy spectra $e_x$ for example test cases $(z^+ = 650, L = \pi/4)$ and $(z^+ = 750, L = \pi/8)$. **The CP-MLP shows finer details on the edge of elements (adjacent squares)**, showing a better prediction of high-order coefficients. The observation is proved by a richer high $k_x$ energy spectra in the right plot.

unknowns *at every time instant*. This includes eight variables at each discretized cell: $\mathbf{q} = [p, u, v, T, Y_{CH4}, Y_{O2}, Y_{H2O}, Y_{CO4}]^T$, where $p$ is the pressure, $u$ and $v$ are the $x$ and $y$ velocity components, $T$ is the temperature and $\{Y_{CH4}, Y_{O2}, Y_{H2O}, Y_{CO4}\}$ are the mass fractions for the chemical species involved in the combustion process. The injector is outlined in Fig. 7, where the oxidizer (O2 diluted in H2O vapor) and fuel (CH4) are injected from two inlets, respectively, into a tube-like combustion chamber in which they mix and react. The products are exhausted through an outlet. A probe monitor is placed inside the physics-intensive area, which is also marked in the figure. The strong instabilities in the simulation is triggered by a strong 2000 Hz pressure perturbation at the outlet. Fig. 7 shows the responses for $p$ and $T$ at the probe. It should be noted that, although the pressure perturbation at the outlet is periodic, the upstream behavior is affected by complex coupled physics and is not as periodic, especially for other variables such as $T$. Fig. 8 shows the graph generated following the method in Sec. 2.1, where special nodes and edges, as well as irregular local structures, are provided in zoomed-in views. Two groups of ghost edges are used, corresponding to two types of wall boundary conditions in the simulation: no-slip and symmetry, respectively.

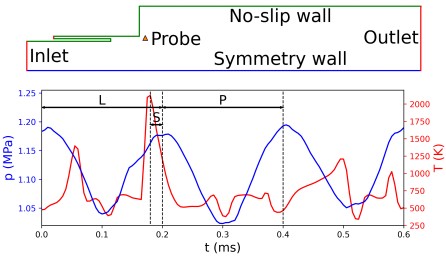

Figure 7: Injector outline and probed response for $p$ and $T$. Orange marker: probe location. L/S: long (0.2 s)/short (0.02 s) training period (0.2 s); P: prediction period (0.2 s).

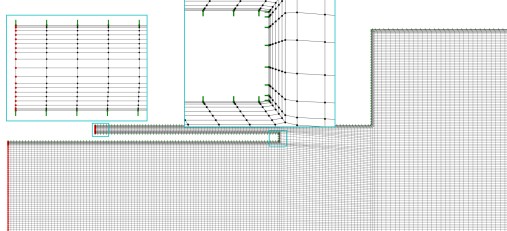

Figure 8: Graph details. Black dots: standard nodes; black lines: standard edges; red dots: inlet/outlet nodes; green/blue lines: two groups of ghost edges (extruded for visualization).

In this experiment, we attempt to predict the future states of $\mathbf{q}$ using the CP-GNet introduced in Sec. 2.1. Two CP-GNets of two different depths, with a 5-block and a 10-block processor respectively, are tested. Both CP-GNets work with an encoded node feature size of 36, and an encoded edge feature size of 4. The baseline model for comparison replaces all CP layers with standard dense layers of 128 units. More specifically, after the replacement, the layers taking node features as conditional parameters will retain the original input. The layers originally taking edge features as conditional parameters will take a concatenation of the original inputs and the edge features as the new input. The non-CP model is referred to as the GNet. GNets, with a 10-block and a 15-block processor respectively, are studied.

The simulation results sampled at a time interval of $5 \times 10^{-4}$ ms are used as the ground truth. Tests are conducted on two different lengths of training data. The long period consists of 400 steps, spanning 0.2 ms, the last 10% of which is used as the short training period. Thus, both periods end at the same point, and rollout prediction is carried out from the end of training for another 0.2 ms. These periods are illustrated in Fig. 7. For simplicity, we add the number of processor blocks and L

(long) or S (short) as suffixes to the model names to distinguish them. For example, "CP-GNet10L" refers to the CP-GNet with 10 processor blocks trained on the long period. The predictions for 4 representative variables, $p, u, T, Y_{CH4}$, from the two deeper models, CP-GNet10L and GNet15L, are visualized in Fig. 9 at 4 steps evenly spanned over the prediction period. The probed results are also plotted, which also covers the other models tested. It is notable that a small phase shift in the resolved structures can cause a high level of deviation in the probe measurements, and thus the flow field contours should be viewed as broader indicators of the performance.

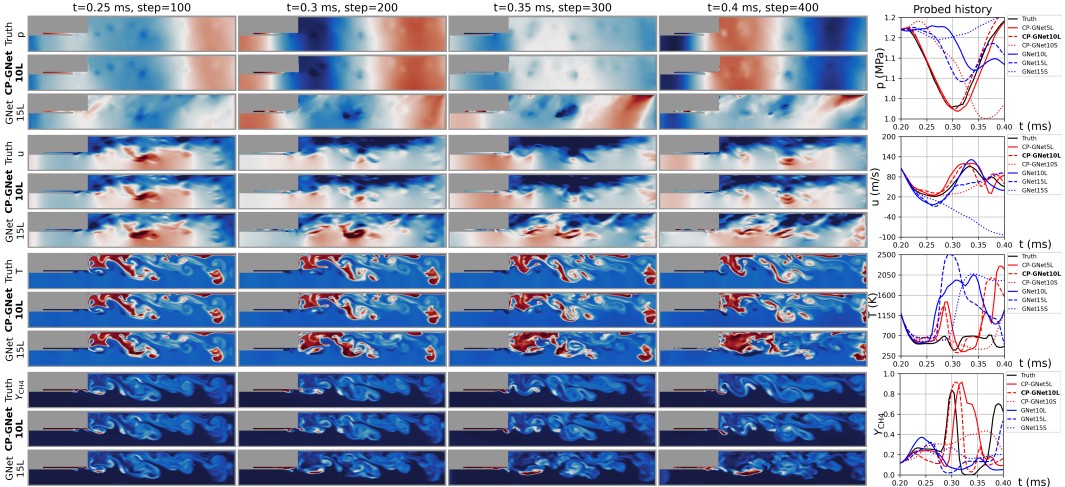

Figure 9: Predicted reacting flow. From top to bottom: pressure $p$, velocity $u$, temperature $T$, mass fraction $Y_{CH4}$. **CP-GNET maintains a high level of accuracy over multiple scales of mesh resolution and near complex geometry boundaries.**

It is seen that the CP-GNET predicts the evolution of the reacting flow accurately over hundreds of prediction steps. In comparison, the non-CP model deviates quickly from the ground truth within 100 steps. Even with a smaller model (CP-GNet5L, 1.3M parameters), or a small fraction of training data (CP-GNet10S), the CP models still show comparable or even better performances compared with the largest baseline (GNet15L, 1.8M parameters). There is no significant difference in the level of error across the predicted field from our model, in spite of the vast changes in mesh density and distortion, whereas the GNets clearly suffer from more errors around the inner corners, where the mesh is the most irregular. This shows that, by combining CP with graph, discretization information can be efficiently processed. The proposed boundary treatment is also proven successful even in such a complex case with multiple types of boundaries (see Appendix A.3.3 for results without ghost edges).

## 5   Summary

This work draws inspiration from discretized numerical methods, and generalizes the idea of conditional parametrization for mesh-based models. Conditionally-parameterized networks can flexibly incorporate physical quantities as well as numerical discretization information into trainable weights, leading to efficient learning of high-order and unstructured features. Drop-in modifications are demonstrated on different architectures for several important tasks related to mesh-based modeling of physical systems. Considerable performance improvements are achieved in the numerical tests compared with the traditional counterparts. In the coarse-graining and super-resolution tasks, a small network with a simple CP-Dense layer is capable of stabilizing or improving numerical solutions. In a test of future state prediction of a rocket injector, the CP-GNet is shown to be capable of predicting the flow with a complex combustion process for a few hundred steps on an irregular mesh. Although a direct CP modification will cause a linear increase in the number of parameters w.r.t. the chosen parameter, such an increase can be compensated by reducing the size of the latent vectors. Indeed, the CP-GNet is more efficient than the non-CP variant with only a fraction of the training data or with a more shallow architecture. In the appendix, we compare the CP-GNet with

the MeshGraphNet on two flow simulation tasks. Overall, the proposed architecture improves the potential for incorporating physical intuition as well as knowledge of numerical discretization.

## Acknowledgments

J.X and K.D acknowledge support from the Air Force under the Center of Excellence grant titled *Multi-Fidelity Modeling of Rocket Combustor Dynamics*. A.P. is supported by NASA under the grant #80NSSC18M0149. We thank Alvaro Sanchez-Gonzalez and Peter Battaglia for valuable advice on training noise injection for robust prediction.

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
