# A    Supplemental Details

## A.1    Data Generation

### A.1.1    Closure modeling

For the present case, the initial condition is given by:

$$u(x,0) = \sum_{k=1}^{8} \sqrt{2E(k)} \sin(kx + \beta_k), \tag{9}$$

where for each $k$, $\beta_k \sim \mathcal{U}(-\pi, \pi)$, and $E(k) = \max(k, 5)^{-5/3}$. Other choices of parameters include domain length $L = 2\pi$, viscosity $\nu = 0.01$.

The 2048 mesh point high-resolution solution is generated using the Fourier-Galerkin spectral method [41] with the 4th order Runge-Kutta method for time stepping. From the box-filtered initial condition, the 32-point low-resolution solution is conducted using central differencing for the spatial derivatives. This choice does not introduce additional artificial viscosity; thus, the solution without closure is naturally unstable. The high-resolution is computed at a small time-step, yet is down-sampled temporally at an interval equal to the low-resolution time step size $\Delta t$=0.0075 s.

In this setting, $u$ can be regarded as fully resolved, thus the numerical residual $r(u)$, defined in Eq. (10), is zero.

$$r(u) = -u \frac{\partial u}{\partial x} + \nu \frac{\partial^2 u}{\partial x^2} - \frac{\partial u}{\partial t}. \tag{10}$$

However, the same does not hold for $\bar{u}$. The ground truth for the closure term completely compensates the non-zero residual, i.e. $\mathcal{C}^* = -r(\bar{u})$.

### A.1.2    Super-resolution

The snapshots for DG projection in this test are sliced from a public dataset [38] for DNS solution for a channel flow at a friction Reynolds number $Re_\tau = \frac{u_\tau h}{2\nu} \approx 950$, where $h$ is the channel height, $u_\tau = \sqrt{\tau/\rho}$ is the wall-friction velocity, defined on the averaged wall-friction $\tau$ and the density $\rho$.

The slices are selected at different normalized wall-distances $z^+ = zu_\tau/\nu$, where $z$ is the distance between the plane to the closer wall.

### A.1.3    Rocket engine injector

The simulation for the public dataset [39] is performed using the finite-volume based General Equation and Mesh Solver (GEMS) [42]. 6 ms of flow is simulated in total at a time interval of $1 \times 10^{-4}$ ms. In our study, the data is downsampled to an interval of $5 \times 10^{-4}$ ms.

## A.2    Network Training

### A.2.1    Hyperparameters

All models are trained with the Adam optimizer. Other training hyperparameters are summarized in Table. 3. For the closure models, the inputs $\mathbf{q}$, $\bar{u}$, and the output $\mathcal{C}$ are normalized by their respective maximum absolute values. No additional scaling is used in the super-resolution task. For the reacting flow simulation task, the different variables in the input $\mathbf{q}$ are normalized to the same order of magnitude. The scaling coefficients are $C_p = 5 \times 10^5, C_{u,v} = 200, C_T = 2500, C_Y = 1$. For the output $\Delta \mathbf{q}$, the scaling coefficients are multiplied by an additional factor $C_\Delta = 0.01$.

### A.2.2    Training noise

We follow the training noise injection strategy as in Refs. [34, 32] to improve the robustness of prediction in the reacting flow simulation task. At the beginning of each training epoch, normally distributed noise $\epsilon \sim \mathcal{N}(0, 0.0013^2)$ is added to the normalized inputs. The variance is selected based on the level of error in the prediction for one step at a time instance away from the training

Table 3: Training hyperparameters.

| Test case | | Batch size | Learning rate | Number of epochs |
|---|---|---|---|---|
| Closure modeling | | 1 | 0.001 | 300 |
| Super-resolution | | 128 | 0.001 | 100 |
| Reacting flow | GNet-S/CP-GNet-S | 1 | 0.002 | 500 |
| | GNet-L/CP-GNet-L | 1 | 0.002 | 100 |

period. The source of this noise is assumed to be from the previous prediction step. The error is supposed to be compensated in the current prediction step; therefore, the noise is subtracted from the target output $\Delta \mathbf{q}$ after being scaled by $C_\Delta$.

### A.3 Additional Analysis

#### A.3.1 Closure modeling

The comparison between CP-CNN and CNN is repeated on 4 other low resolution meshes of different sizes $n_x = \{24, 64, 128, 256\}$. The average MAE for the online computation for $\bar{u}$, and the offline single-step computation for $\mathcal{C}$ from the training IC is plotted in Fig. 10, along with the results for $n_x = 32$ from Sec. 4.1. The CP-CNN outperforms the CNN on all meshes. Moreover, the CNN closure is unstable at the most coarse mesh, $n_x = 24$, whereas the CP-CNN is stable.

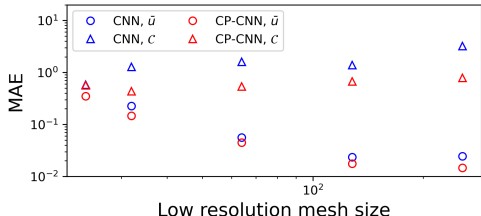

Figure 10: Average MAE for $\bar{u}$ (online) $\mathcal{C}$ (offline) under different low resolution mesh sizes. **The CNN model blows up at $n_x = 24$. The CP-CNN outperforms the CNN on all meshes.**

#### A.3.2 Super-resolution

The definition for the energy spectra used in Sec. 4.2 is given by:

$$e_x(k_x) = \frac{1}{\pi} \int_{-\infty}^{\infty} \langle u(x_0, y_0) u(x_0 + x, y_0) \rangle \, e^{-ik_x x} dx, \tag{11}$$

$$e_y(k_y) = \frac{1}{\pi} \int_{-\infty}^{\infty} \langle u(x_0, y_0) u(x_0, y_0 + y) \rangle \, e^{-ik_y y} dy, \tag{12}$$

where $\langle \cdot \rangle$ denotes the average over homogeneous directions, which is the entire plane in this case. Similar to the power spectral density for a time series that describes the energy distribution over different frequencies, the energy spectra describes the energy distribution of a spatial field over different wave-numbers $k = 2\pi/\lambda$, $\lambda$ being the wavelength.

#### A.3.3 Rocket engine injector

**Additional variables.** As a supplement for Fig. 9, predictions for the rest variables, $v$, $Y_{O2}, Y_{H2O}, Y_{CO2}$, are provided in Fig. 11. They further validate the conclusions in Sec. 4.3.

**Prediction for a distanced and longer period.** To test generalization performance, the CP-GNet10L model is used to perform prediction at a new time instance $t = 2$ ms, which is away from the training period. The predicted period is also doubled to 0.4 ms. The results are provided in Fig. 12. For the first 0.15 ms, a similar level of accuracy is obtained compared with the previous run that is appended to the end of the training period. However, the prediction is unstable in the long term, illustrated by scattered extreme values in the contours. The authors acknowledge that long-time stability guarantees is a major limitation of the current - and existing - work in the domain of data-driven flow predictions.

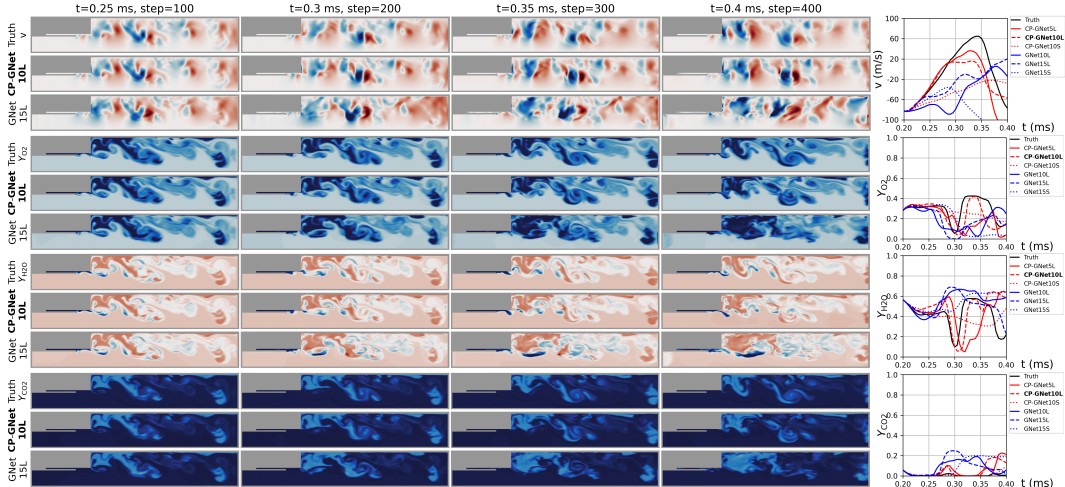

Figure 11: Predictions for the rest variables $v$, $Y_{O2}$, $Y_{H2O}$, $Y_{CO2}$.

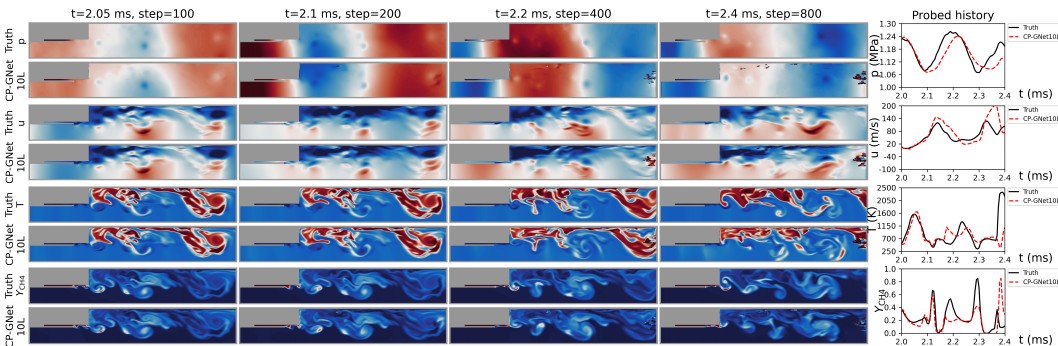

Figure 12: Predictions from a new time instance $t = 2$ ms for a longer period of 0.4 ms. The initial 0.15 ms shows a similar level of accuracy as the appended case. The prediction becomes unstable in the long term, illustrated by scattered extreme values in the contours.

**Results without the boundary treatment.** The CP-GNet10L model is re-trained on a graph without ghost edges for the wall boundaries. The prediction is again started at the end of the training ($t = 0.2$ ms). The results for $p$ and $u$ at the last prediction step $t = 0.4$ ms are shown in Fig. 13. The accumulation of error is clearly visible in several near-wall regions, which validates our proposed boundary treatment.

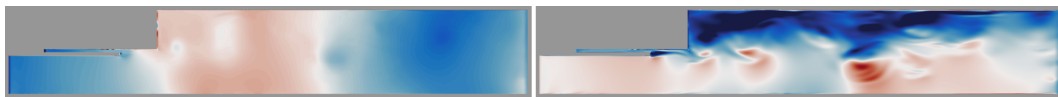

Figure 13: Results on a graph without ghost edges for $p$ (left) and $u$ (right). The accumulation of error in the near-wall regions is clearly visible.

**Timing.** The prediction for 0.1 ms of flow with the CP-GNet10L model takes 53 seconds on one Nvidia RTX A6000 GPU, or 599 seconds on 40 CPU cores. In comparison, the original simulation for 6 ms of flow takes approximately 1200 CPU hours [43]. Due to different hardware configurations, no direct comparison can be made. However, we can safely estimate a 2.5x∼3x speedup on CPUs, and a 25x∼30x speedup when a GPU is utilized.

# B  Generalizable & Exact Fitting with CP

In this appendix we demonstrate - as a proof-of-concept problem - how discretized PDE terms can be fitted exactly with simple CP layers in the solution of the 2D advection-diffusion equation. This experiment is performed in the limit of extremely sparse data snapshots. With periodic boundary conditions, the governing equations are:

$$
\begin{aligned}
&\frac{\partial u(x,y,t)}{\partial t} + \mathbf{a}\nabla u(x,y,t) - \nu\nabla^2 u(x,y,t) = 0,\\
&x \in [0,W], y \in [0,H], t \in [0,T],\\
&u(0,y,t) = u(W,y,t), u(x,0,t) = u(x,H,t),
\end{aligned}
\tag{13}
$$

where $\mathbf{a} = [a_x, a_y]^T$ is the advection velocity vector. The initial condition (IC) has a rectangular frame with $u = 1$ in the center, and $u = 0$ elsewhere, as shown in Fig. 14.

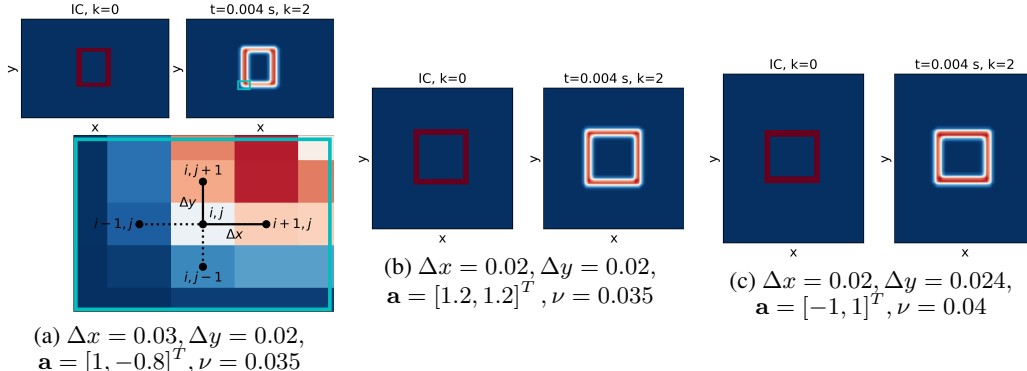

(a) $\Delta x = 0.03, \Delta y = 0.02,$
$\mathbf{a} = [1, -0.8]^T, \nu = 0.035$

(b) $\Delta x = 0.02, \Delta y = 0.02,$
$\mathbf{a} = [1.2, 1.2]^T, \nu = 0.035$

(c) $\Delta x = 0.02, \Delta y = 0.024,$
$\mathbf{a} = [-1, 1]^T, \nu = 0.04$

Figure 14: Training data. **Each case consists of only the IC and two solution steps.**

The first-order upwind discretization of Eq. (13) is given by:

$$
\begin{aligned}
&\frac{\Delta u_{i,j}^k}{\Delta t} + \frac{a_x + |a_x|}{2\Delta x}(u_{i,j}^k - u_{i-1,j}^k) + \frac{a_x - |a_x|}{2\Delta x}(u_{i+1,j}^k - u_{i1,j}^k) + \frac{a_y + |a_y|}{2\Delta y}(u_{i,j}^k - u_{i,j-1}^k)\\
&+ \frac{a_y - |a_y|}{2\Delta y}(u_{i,j+1}^k - u_{i,j}^k) - \nu\frac{u_{i-1,j}^k - 2u_{i,j}^k + u_{i+1,j}^k}{\Delta x^2} - \nu\frac{u_{i,j-1}^k - 2u_{i,j}^k + u_{i,j+1}^k}{\Delta y^2} = 0,
\end{aligned}
\tag{14}
$$

where $i$ and $j$ are the grid indices in the $x$ and $y$ directions, respectively, and $\Delta x$ and $\Delta y$ are the distances between grid points, as illustrated in Fig. 14a.

Inspired by Eq. (14), a neural network model is constructed using a dense layer and two 2D CP-Convolution (CP-Conv) layers in the form:

$$
\begin{aligned}
\mathbf{h} &= \mathrm{ReLU}(\mathbf{W}_1\mathbf{a}),\\
\Delta\mathbf{u}^k &= \left\langle \mathbf{W}_2, \frac{\Delta t}{\Delta x}\mathbf{h}\right\rangle * \mathbf{u}^k + \left(\left\langle \mathbf{W}_2, \frac{\Delta t}{\Delta y}\mathbf{h}\right\rangle * (\mathbf{u}^k)^T\right)^T\\
&+ (\mathbf{W}_3\frac{\nu\Delta t}{\Delta x^2}) * \mathbf{u}^k + ((\mathbf{W}_3\frac{\nu\Delta t}{\Delta y^2}) * (\mathbf{u}^k)^T)^T,
\end{aligned}
\tag{15}
$$

where $\mathbf{W}_1 \in \mathbb{R}^{2\times2}$ is the weight for the dense layer, and $\mathbf{h}$ is the hidden output. $\mathbf{W}_2 \in \mathbb{R}^{3\times1\times1\times2}$ is weight for the first CP-Conv kernel, taking $\frac{\Delta t}{\Delta x}\mathbf{h}$ or $\frac{\Delta t}{\Delta y}\mathbf{h}$ as the condition parameter. $\mathbf{W}_3 \in \mathbb{R}^{3\times1\times1\times1}$ is weight for the second CP-Conv kernel, taking $\frac{\nu\Delta t}{\Delta x^2}$ or $\frac{\nu\Delta t}{\Delta y^2}$ as the condition parameter. $(\cdot * \cdot)$ denotes the convolution operation.

In common deep learning applications, a large amount of training data is required to train the model sufficiently, and to avoid overfitting. In this test case, however, we assess the ability of the model to approximate the truth at a machine precision level using limited data. As a demonstration, we use

Table 4: Training hyperparameters.

| Number of snapshots | 6 |
|---|---|
| Grid points per snapshot | $51 \times 51$ |
| Batch size | 1 |
| Initial learning rate | 0.1 |
| Final learning rate | 0.0003 |
| Number of epochs | 10000 |

only 3 sets of 2-step training data snapshots, each for a different set of parameters $\{\Delta x, \Delta y, \mathbf{a}, \nu\}$ to train a model represented by Eq. (15). Each set only has solutions for two time steps beyond the initial condition. All of the training cases are present in Fig. 14. The model is trained with Adam optimizer with the training hyperparameters listed in Table 4.

The weights learnt [2] are:

$$\mathbf{W}_1 = \begin{bmatrix} 0.33237486 & 0 \\ 0 & -0.5253752 \end{bmatrix}, \mathbf{W}_2 = \begin{bmatrix} 3.00869074 & -3.00892553 & -8.69012577 \times 10^{-5} \\ 2.38949641 \times 10^{-5} & -1.90361486 & 1.90334544 \end{bmatrix}^T,$$

$$\mathbf{W}_3 = [0.99999235, -1.99994342, 1.00001345]^T.$$

Indeed when the weights are substituted into Eq. (15), we recover Eq. (14) to 4-decimal-point precision, as the ideal weight combination is

$$\mathbf{W}_1 = \begin{bmatrix} 0.5c_1 & 0 \\ 0 & -0.5c_2 \end{bmatrix}, \mathbf{W}_2 = \begin{bmatrix} 1/c_1 & -1/c_1 & 0 \\ 0 & -1/c_2 & 1/c_2 \end{bmatrix}^T, \mathbf{W}_3 = [1, -2, 1]^T,$$

where $c_1, c_2 \in \mathbb{R}_{\neq 0}$.

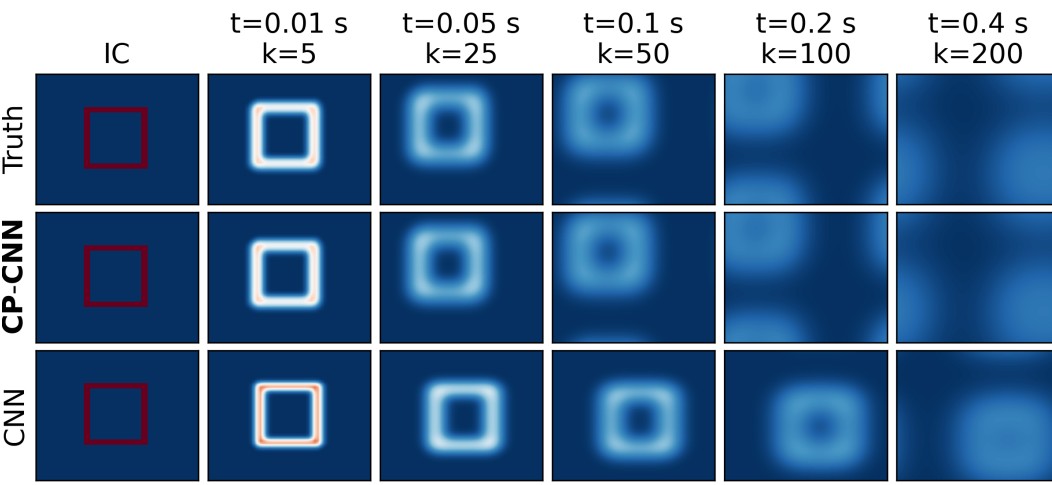

Figure 15: Prediction results. CP-CNN fits the discretized model exactly. The CNN is highly inaccurate.

The model is used to perform rollout prediction for 200 steps at a new initial condition and set of parameters outside the training range, $\Delta x = 0.02, \Delta y = 0.016, \mathbf{a} = [-1.5, 1.5]^T, \nu = 0.02$. The result is present in Fig. 15. The L1 error is less than $1e-4$.

In comparison, a CNN with the CP-Conv layers in Eq. (15) replaced by standard convolution layers are applied to the same task. It can be seen from Fig. 15 that the CNN can represent neither the advection nor the diffusion correctly.

It is thus clear that for this class of problems, the utility of conventional deep learning is questionable. The authors note that, in practical and more complex applications, an exact fitting will not be achievable even with CP. However, as an idealized demonstration, this test serves the purpose to show how CP networks can represent functional relations between parameters and variables to reduce the training effort for certain terms, improve accuracy, and offer generalizable predictions.

---

[2] $\mathbf{W}_2$ and $\mathbf{W}_3$ squeezed for simplicity

Table 5: Averaged inference time and RMSE for reacting flow.

| Model | Time/step ms | RMSE 1-step $\times 10^{-3}$ | RMSE rollout-50 $\times 10^{-3}$ | RMSE rollout-all $\times 10^{-3}$ |
|---|---|---|---|---|
| CP-GNet | 261 | 0.29 | 6.8 | 46.1 |
| 128-unit MeshGraphNets | 203 | 0.42 | 10.4 | 62.8 |
| 256-unit MeshGraphNets | 296 | 0.41 | 10.8 | 58.4 |

## C  Comparison against MeshGraphNets

Although both methods are designed for mesh-based simulations using graphs, the CP-GNet is inspired by the finite volume method, leading to different graph representations. In addition to the conditional parametrization, the three main differences between the two approaches are: 1) The edges are unidirectional between nodes for the CP-GNet v.s. bidirectional for the MeshGraphNets; 2) Graph nodes are located at mesh cell centers for the CP-GNet v.s. mesh vertices for the MeshGraphNets; 3) The boundary conditions are implemented by adding ghost edges for the CP-GNet v.s. distinguishing node labels for the MeshGraphNets. Due to these differences, a strict direct comparison between the two methods becomes infeasible. To provide a meaningful comparison, both models are tested on a FVM dataset (reacting flow) and a FEM dataset ( incompressible flow over a cylinder). Minimal adjustments to the models/input features are made accordingly for migration purposes.

### C.1  Reacting flow

The long-training-period experiment setting from Sec. 4.3 is used. To apply the MeshGraphNets, one-hot labels distinguishing fluid cells and different types (symmetric wall/no-slip wall/inlet/outlet) of boundary cells, as well as the cell volumes are added to the node features. Face areas between cells are added to the edge features. Moreover, we also tested a wider version of the MeshGraphnets, with the default 128-unit MLPs replaced by 256-unit ones, due to the complex reaction physics in this task. Both MeshGraphNets are trained using the same training hyper-parameters as the CP-GNet10L model from Sec. 4.3, and compared with the latter.

Evaluations are again performed on the representative variables $p, u, T, Y_{\text{CH4}}$. The predicted flow fields are visualized in Fig. 16. It can be seen that the CP-GNet is visually closer to the truth, especially in the phases of the probed peaks. The averaged inference time and RMSE for the normalized variables (with mean subtracted, divided by standard deviation) at different rollout steps is reported in Table 5. The present model provides a lower RMSE throughout the prediction.

### C.2  Incompressible flow over a cylinder

For this experiment, the setting for incompressible flow over a cylinder from the MeshGraphNets [34] is adopted. The task is to predict the 2D velocity components stored on the vertices on irregular triangular meshes. The data includes 1000 training trajectories and 100 testing ones, each with 600 steps. In the original setting, 10 million training steps are used, which takes several days on a single GPU. In this work, an additional comparison is performed after 2.5 million training steps to evaluate the training efficiency of the models.

The MeshGraphNets results are generated using the official code. The CP-GNet model is migrated to the same pipeline, with two minor adjustments made: 1) A two-layer MLP is added to the node encoder to process the node label; The network width is increased from 36 to 64 to process the additional features. 2) The CP-Dense layer in the "source term" section in the processor is removed as there is no chemical reaction taking place in this test.

The averaged inference time and RMSE for the testing trajectories are summarized in Table 6. It should be noted that a single A6000 GPU is used in our tests, and a V100 is used in [34], and noticeably different numbers are reported. At a smaller number of training steps (2.5 M), our model provides a higher single-step RMSE, yet a lower long-rollout RMSE. Compared with the previous test, the efficiency of our model is largely affected by the additional processing of node labels and a larger network width, running significantly slower than the MeshGraphNets. The predicted final steps for 5 randomly selected testing trajectories are visualized in Fig. 18. At 2.5 M training steps,

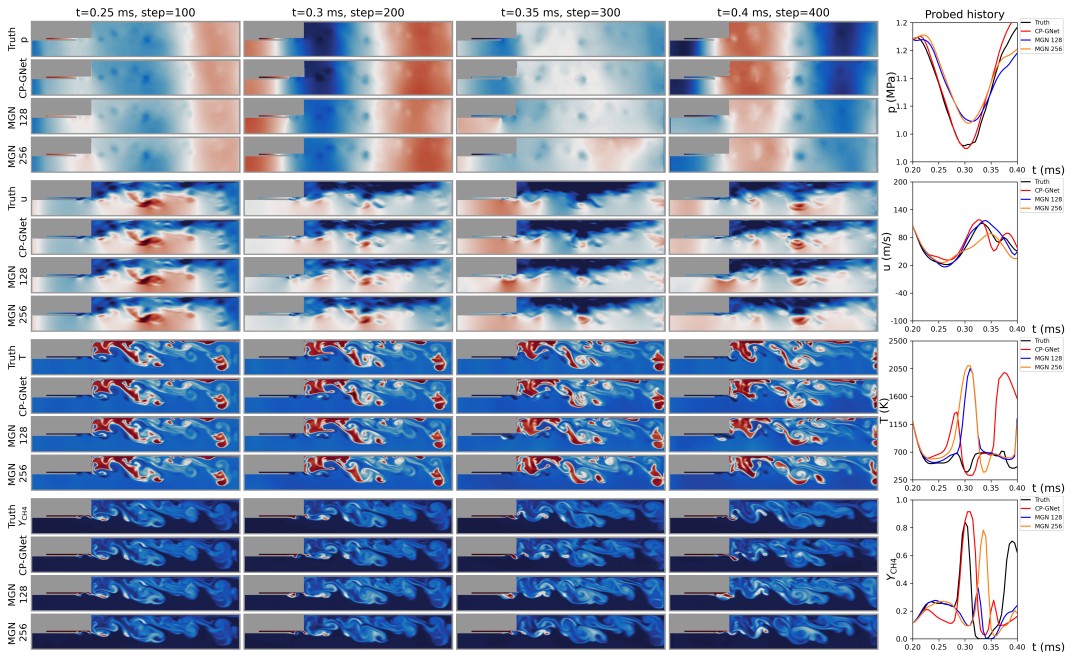

Figure 16: Predicted reacting flow. For each variable from top to bottom: ground truth, CP-GNet, 128-unit MeshGraphNets (MGN 128), 256-unit MeshGraphNets (MGN 256).

Table 6: Averaged inference time and RMSE for flow over cylinder.

| Model (training steps) | Time/step ms | RMSE 1-step $\times 10^{-3}$ | RMSE rollout-50 $\times 10^{-3}$ | RMSE rollout-all $\times 10^{-3}$ |
|---|---|---|---|---|
| CP-GNet (2.5 M) | 16 | 3.3 | 12.4 | 62.5 |
| CP-GNet (10 M) | 16 | 2.8 | 9.9 | 54.0 |
| MeshGraphNets (2.5 M, tested) | 9 | 2.1 | 8.7 | 68.5 |
| MeshGraphNets (10 M, tested) | 9 | 1.9 | 6.9 | 50.1 |
| MeshGraphNets (10 M, reported [34]) | 21 | $2.34 \pm 0.12$ | $6.3 \pm 0.7$ | $40.88 \pm 7.2$ |

the CP-GNet performs better in two unsteady cases (trajectories #8 and #17). However, it over-predicts the velocity magnitude in the two steady cases (trajectories #32 and #72). Meanwhile, the MeshGraphNet under-predicts in one of them (trajectory #72). At the larger number of training steps (10 M), the MeshGraphNets shows a lower error across the prediction period, with the gap between the two models decreasing with the number of rollout steps. The differences in the final step predictions become less significant.

It should be pointed out that between this and the previous test problems, there are many factors that can lead to changes in model performances, including the type of the ground truth solver and data (cell-centered FVM vs. vertex-centered FEM), the number of variables (8 vs. 2), the implement of boundary conditions (ghost edges vs. node labels). Based on these specific results, one cannot make a definitive statement on the relative merits of each of the two approaches.

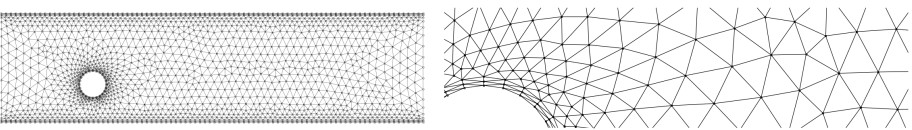

Figure 17: Example irregular mesh for the flow over cylinder with a zoomed-in view on the right

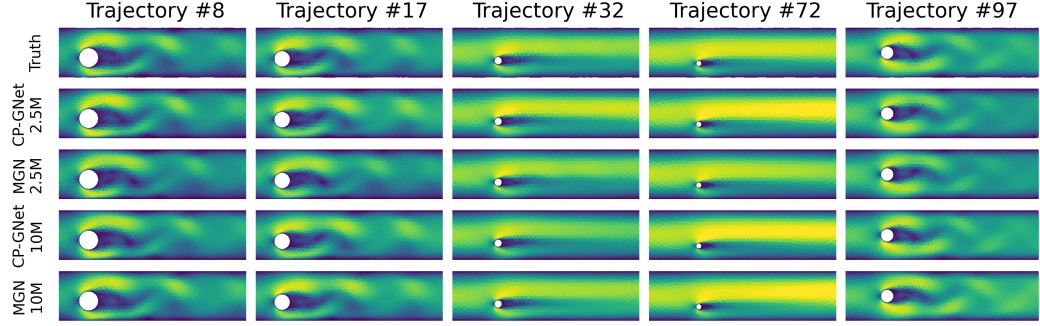

Figure 18: Velocity magnitude for the last step in the rollout prediction for random testing trajectories. From top to bottom: ground truth, CP-GNet, MeshGraphNets (MGN).