# OpenReview forum: "Conditionally Parameterized, Discretization-Aware Neural Networks for Mesh-Based Modeling of Physical Systems"
_NeurIPS.cc/2021/Conference — NeurIPS 2021 Poster_

### Official Review · Reviewer_t6t9 · 2021-07-06

**Rating:** 6
**Confidence:** 4

**Summary:**

The paper presents a new "conditionally parametrized" NN layer for learned physics models, and demonstrates higher prediction accuracy on several physics prediction tasks by injecting this new layer into CNN and GNN-based models.

**Ethical Concerns:**

None.

**Limitations And Societal Impact:**

No issues here.

**Main Review:**

This is an interesting paper; the authors showed results on a range of interesting & complex physics prediction tasks, and the method seems to perform well compared to state-of-the-art baselines. I however wish the paper did a more thorough analysis as to why the model performs better, and provide the relevant ablation experiments.

The method is first introduced and motivated as a type of hyper network, i.e. predicting the weights using a different network. I'm not sure this is the most useful framing; there is no separation between the network and hyper network, the model is trained end-to-end, and in most cases conditioning p and input u as set to be the same vector.
Later in the paper it is mentioned at that the CP layer performs well because it explicitly contains products of u, and thus can better model the nonlinear PDE terms. This intuition makes much more sense to me; and I think in spirit this paper is much closer to multiplicative conditioning methods such as e.g. FiLM [Perez et al 2017] than hyper networks.
If this is the case, I'd be very interested to see how other, less exotic form of multiplicative layers would perform-- e.g. f(g(u) * h(p)) (with * being element-wise multiplication), forms of FiLM, or simply providing the matrix u'u as an input to the first network layer. This is a bit of a missed opportunity, as it could not only provide more insight in where the performance gains really come from, but also allow to construct a layer which can use wider latents without explosion in parameters (as CP-Dense is O(n^3)).

The second claimed contribution ("discretization-aware") of the method over other MPNN models is explicitly weighting the incoming edge messages with the FVM flux weight w_ij. It should probably be mentioned that this only works with cell-centered data from a FVM simulation, and wouldn't work on the particle/FEM datasets used in related work. More importantly, there needs to be an ablation to determine how much of the performance gain is due to this, vs. the CP-* layers. This should be very easy to do; remove the weighting in eq.5 and instead provide the unnormalized edge vector x_i-x_j (and perhaps edge length) as an edge features as in [32, 34].

I really enjoyed the range of complex physics tasks studied in the results section. However, the comparison often were not performed in an apple-to-apples manner, e.g. replacing a Dense layer (O(n^2) parameters, 1 nonlinearity) with a CP-Dense layer (O(n^3) parameters, 2 nonlinearities). These comparisons really need to be same-capacity, i.e. more & wider Dense layers.
This is particularly obvious in 4.1: it's really not that surprising that a 2-layer CNN won't be very good at capturing nonlinearities, but perhaps a deeper network with matched capacity might.
The other observation on results is that the test sets are generally in-domain; and the only generalization test in the appendix showed stability issues. This is slightly concerning, as [32, 34] demonstrate stable very long-term rollouts and strong OOD generalization, so it'd be good to a) show baseline comparison for this generalization test to determine if this behavior is specific to the CP model, and b) investigate the cause of the instabilities.

---
Minor comments:
- Line 75: wrong equation reference
- I assume DDP has a Dense(16), not CP-Dense(16) layer?
- How exactly does the CP layer in CP-CNN work?
- Tab:1 is highlighted incorrectly, DDP has lower c_avg/c_max

**Time Spent Reviewing:**

4

---

> ### Author Response · Authors · 2021-08-10
> **Clarifications**
>
> Thank you for reviewing our work carefully. Your detailed comments are quite insightful. We would like to provide clarifications and justifications for some of our  choices.
>
> On the framing of the introduction, we agree that in the more complex networks, the relation to the hypernetwork becomes vague, because multiple stacks of CP layers are trained jointly in an end-to-end manner. We agree that an internal stack would be beneficial. An additional discussion on the analogy to FiLM and other mechanisms such as gating and attention would be a good contribution. A major difference between FiLM and our method is that the former is typically applied to intermediate/output dimensions as an element-wise multiplication. Our method, in contrast, determines the weight matrix in a tensor product manner, and thus is closer to the hypernetwork, despite differences such as the unusual internal stacks.
>
> The number of trainable parameters for the CP-Dense layer is $O(n_x^2\times n_p)$, assuming $n_x$ is the input size, which is much lower than $O(n_x^3)$ in our applications. For the first two easier tasks, CP with $n_p = 2$ is only applied to one layer in each model, which only doubles the number of weights in that single layer. In our preliminary parameter tuning runs for the baselines, we made sure that the size of the original baseline is large enough for the task such that even doubling it in a naive manner will not improve the accuracy noticeably. A parameter study for the baselines can be included into the appendix to address this concern. Furthermore, we do agree that the choice of a 2-layer CNN may be too simple, and the DDP model, which is a MLP instead of CNN, cannot fully serve the purpose of showing the impact of CP at different network depths. This could also be addressed in the parameter study. For the third task, in most layers, $n_p$ is much smaller than $n_x$, e.g., in the processor, $n_p=4$, which is the size of the encoded edge feature, whereas $n_x=36$. One exception is the encoder of the CP-GNet, where the input feature is also taken as the parameter therefore $n_p=n_x$. However in this case, $n_x$ itself is shrunk, and the CP and non-CP models still have a similar number of parameters (e.g. 1.28M for CP-GNet5, 1.22M for GNet10). This will be made explicit in the  updated version.
>
> We have also performed more tests on exact MeshGraphNets models. In general, we find that it under-performs compared to our baseline model.We believe that this is because of the following differences between our approach and MeshGraphNets: 1) the computation of source term in reaction is very challenging for pure message-passing type processors, but can be taken care of by our "source term section" (shown in Fig 1) of the processor. 2) The original MeshGraphNets updates a large set of edge features through processor blocks, which is hard to train in our setting of limited training data. We would behappy to report the results in the appendix, emphasizing that the performance difference could be highly dependent on the problem and the training resources.
>
> On the concern of long-term rollout prediction, we have also prepared results for the classic (and less challenging) cylinder flow problem, where the CP-GNet is able to predict stably and accurately for more than 1500 steps. Empirically, reacting flows, as the one we chose, suffers more from numerical instabilities in traditional simulations due to multiple reasons, such as a small characteristic time, large local gradients of temperature, etc. The investigation on them and other causes of instabilities in our setting will compose an important aspect of our future studies. We do hope, however, that the choice of an extremely complex application is not held against us!
>
> We also appreciate the comments on the writing and visualization. Your observations are correct and we will edit these parts accordingly. In the CP-CNN model, the CP only applies to the initial dense layer, and the convolution layer itself is a standard one.

---

> > ### Comment · Reviewer_t6t9 · 2021-08-31
> > **Updated review**
> >
> > Thanks for the response!
> > Yes, if CP-Dense layers are purely used with physical quantities as inputs ("source selection"), then the blow-up in parameters is less bad, but of course this also restricts its use cases. I still think the paper would strongly benefit from comparisons to other forms of multiplicative embeddings-- e.g. the ability to use multiplicative embedding beyond source selection (e.g. on intermediate latents) could be a good thing, and also an ablation with just manually providing products of input quantities as inputs would be good to have as well.
> > And I'd still think there should be an ablation for the impact of including the flux terms.
> >
> > I'm happy that the authors decided to add new baselines comparisons with Meshgraphnets and rollout tests on cylinder flow. While I won't raise my score further due to the concerns in my main review and above, I'm still (weakly) in favor of accepting this paper.

---

> > > ### Author Response · Authors · 2021-09-15
> > > **Updates on comparison with MeshGraphNets and flow over cylinder**
> > >
> > > Thanks for your updated comments! We would like to followup and update you with the comparison we performed against the MeshGraphNets in response to your comments. The comparison is conducted with two tests. The first test is the reacting flow case in Sec 4.3, where we implemented the MeshGraphNets in our pipeline. Node labels distinguishing fluid and different types of boundary cells are added to the node features as required by the MeshGraphNets. Due to the complex physics, we also tested a 256-unit MeshGraphNet model, which is twice the width of the original 128-unit model. The 1-/50-/400-step rollout RMSE are {$0.29, 6.8, 46.1$}$\times 10^{-3}$ for our model (2.5M parameters), {$0.41, 10.8, 58.4$}$\times 10^{-3}$ for the 256-unit MeshGraphNets (7.1M parameters), and even higher for the original 128-unit model (1.8M parameters). Visualization results are also better for our model.
> > >
> > > The second test is the incompressible flow over a cylinder case from the MeshGraphNets work. We implemented our model with their official open-sourced code set and data. The 1-/50-/600-step rollout RMSE for the velocity components are {$3.3, 12.4, 62.5$}$\times 10^{-3}$ for our model, and {$2.1, 8.7, 68.5$}$\times 10^{-3}$ for the MeshGraphNets. From visualizations for randomly selected trajectories, we observed that our model shows clear advantages in the unsteady ones (with oscillating downstream flow), whereas the MeshGraphNets perform slightly better in the steady ones (with a straight downstream flow).

---

### Official Review · Reviewer_hEWt · 2021-07-12

**Rating:** 8
**Confidence:** 4

**Summary:**

The authors of the paper are adding a  parametrized (by input data) layer to perform a renormalisation step from a small scale PDE to larger scale system. This requires a definition of a closure term that modifies the equation at the larger scale to match the behaviour at the a more accurate, smaller scale. This is shown to significantly reduce the required computing resources, while still retaining high accuracy.  More, the system is defined on graph convolutional network that is applicable to complex mesh structures that are required for proper description of many real world problems.

The authors introduce a simplified form of the attention mechanism that is missing the value vectors and softmax portions, but gives a possibility to introduce a degree of  non-linearity to the activation, besides the activation function. The expectations is that this will help the network to reach accuracy with fewer computations. This is  validated in the  CFD cases considered in the paper.

**Ethical Concerns:**

The paper domain does not bring up ethical questions. Some use cases in the above mentioned domain may be considered. My personal opinion is that this is technical development that has still some steps to go before "blind" adoptions for crowd and traffic control applications. I do not think an ethical review is needed. The ethical aspects arise in those applications when someone starts utilising the results of this work  for those purposes. This is just to remind the authors that ethical consideration may arise from unexpected uses of algorithms.

Of course, this comment, is valid for all PDE solvers that can be used for in the realm of social science.  I think an ethical review is applicable when there is a paper working in those specific areas explicitly.

**Limitations And Societal Impact:**

No limitations. The code may have social impacts if the results are used for traffic or crowd modelling, and the models would not be accurate enough for example for escape route planning from burning buildings.

**Main Review:**

The authors report significant advances for easing the computational load of turbulent flows using AI methods. It is also done in a manner that requires only limited, practical amount of for training as quite a lot of the architecture is directly defined by the nature of the PDE to be solved.

The text is well, written and understandable, although it requires a bit of domain knowledge on both of the AI side and numerical methods for solving PDEs. All essential parts are explained and appendix provides clarifying information. The authors make a good work in compressing the topic to a concise package. Especially the I like the Figure 1 that describe the used architecture in clear and explicit manner. The related work is well represented, even in terms of comparing performances.

As their first test case, the authors use the 1-d, Naiver stokes equation with viscosity i.e the Burgers equation. The authors do not mention the Courant condition, and how the viscosity used would be related to the real physical value of the viscosity the system represents. Doubling the spatial size, would require a significant reduction of the temporal step. Also, how the stability of the method is followed on irregular meshes with a given time step size. The authors should, perhaps in the Appendix, consider how their scheme is solving these issues. Just looking at nice conforming solution examples does not bring a certitude required from a PDE solver. One-dimensional Burgers equation has analytically expressed solutions that can be used for verification as well. The example on rocket engine injection, using public high quality numerical solution, shows quite impressive performance of the method.

From a point of view of the deep learning, the architecture, and the structure of the new solution is described well, Also the the training data and the training process is clear.

Some interesting points remain unsolved: as far I understand the system trains itself on a solution itself that it will propagate further in time. Hence, one has to start solving the system from a given boundary conditions (and parameters like the viscosity), train the proper parametrised weights for the system (restricted  amount of data needed!) and the the AI solutions brings an efficient prediction of the future values. This does not show a very high degree of generalisation. Could one solve the system with a more extended set of varying conditions, like the changing, for example,  the viscosity (the physics of the system) and test it with viscosity values it has not been trained on before? Of course, the viscosity could be entered in as a conditional parameter.  This would increase the usability of the code, as a more general form of the closure would be already trained and available. I would try to remove the need of a training step in the inference that now seems to be there, at least in the examples shown.





**Time Spent Reviewing:**

5

---

> ### Author Response · Authors · 2021-08-10
> **Clarifications**
>
> Thank you for reviewing our work and providing insightful comments. We hope to address some of your  concerns below.
>
> The CFL number is below 0.4 for all 1D Burgers cases, including the ones in the appendix. Your suggestions on comparisons against an analytical set of solutions and a viscous term would be a beneficial addition to further demonstrate the role of the closure model, which we will add in the appendix. The stability on irregular meshes would however require much more extended study, requiring substantial changes to the baseline model, and more features included into the conditional parameters. For this reason, although on an interesting and valuable track, we believe it would fall beyond the scope of our current work.
>
> On the generalization concern, our design for Appendix B was indeed setup to show that the network makes predictions out of the box without further training on a new system if parameterized appropriately. We are aware that the test case is ideal and a replication on one of the more complex tests would perhaps be more convincing. We would be happy to perform a similar test in a similar setting as the one for Appendix A.3.1, but with a single model trained instead of separate ones.
>
> We also appreciate the ethical and impact concerns you brought up. This made us revisit the scope of the work and consider possible applications that were beyond our thinking at the time this work was initiated.

---

### Official Review · Reviewer_o2jT · 2021-07-16

**Rating:** 6
**Confidence:** 4

**Summary:**

The paper proposes to incorporate the idea of 'conditional parametrization' (CP) in neural network modeling of PDEs. In experiments covering a closure model, superresolution, and time evolution of a multidimensional system, CP is shown to significantly improve results.


**Limitations And Societal Impact:**

The authors do not discuss potential negative impact due to perceived low risk, which seems fine for this type of work.

**Main Review:**

The authors propose that the idea to use a neural network to generate the weights of another neural network is useful in the context of PDE modeling. This concept, referred to as CP in the present work, is not in itself novel -- similar solutions were discussed before in CondConv, hypernetworks or Edge Conditioned Convolution. The specific application to PDEs in general, and PDEs modeled with GNNs in particular, is a contribution of the present work. Prior work is appropriately cited and discussed.

The main claim of why CP is helpful is that "higher order terms and irregular discretizations" can be modeled more efficiently. In the appendix, the authors show that in some simple cases when CP is used and the network is appropriately structured, the exact form of the PDE can be recovered even with very limited training data. CP is then tested on 3 problems of increasing complexity, of which only the last involves non-Euclidean grids and GNNs.

The 1st problem is closure modeling for the coarse-grained Burgers' equation. The baselines are simple CNN and MLP models, which are then extended with CP. CP is shown to keep the solution bounded for all models and significantly improves precision for the CNN.

The 2nd problem is super-resolution in turbulent channel flow snapshots, realized with a discontinuous Galerkin projection. The network is trained to predict high-order basis coefficients from low-order basis coefficients for the node and its neighbors, as well as the local Reynolds number, which is concatenated to the input when an MLP is used and provided as a parameter to the CP-MLP. CP-MLP is visually shown to produce more complex flow patterns, richer energy spectra, and lower energy errors.

The 3rd problem is combustion in a 2d model of rocket engine injector, solved with a 2d FVM on a mesh with ~38k nodes, each containing 8 variables. In this experiment, variants of MeshGraphNets with and without CP are tested, with the CP ones staying significantly closer to the ground truth. No metrics are provided for this experiment, but a visualization of the flow fields together with the measurements from a probe point are sufficient to see the difference.

The paper is easy to read, introduces core concepts in a sensible order, and keeps the right amount of details in the main text. A statement of contributions could be a helpful addition so that it's clear what exactly is new. For instance, ~2 pages are spent introducing the finite volume formulation and CP-GNet, which can be viewed as a CP modification of MeshGraphNets. Some of the claims could be better explained and motivated -- for instance, in the discussion a claimed advantage of CP is that it improves the "potential for incorporating physical intuition as well as knowledge of numerical discretization", but it isn't clear how this was the case in the 3 experiments.

The reported results are interesting and promising, but there are two downsides to the experimental protocols. First, when CP is applied to a network, it increases the number of trainable parameters and the effective depth. This is however not taken into account at all in the first two examples, and partially taken into account in the last example (but without any careful accounting for either parameters or depth). Arguably, a fair baseline for the CP models should have comparable depth and parameter count, and access to the same input information (e.g. DDP in the closure problem doesn't get 'q' as an input).

The second downside is the lack of comparison against externally reported baselines. In the Burgers' equation one could compare against e.g. https://doi.org/10.1073/pnas.1814058116. The MeshGraphNets paper covers a number of nontrivial cases -- given the focus on fluid flow in the present work, the airfoil or cylinder flow examples would be natural settings to test before introducing the more complex combustion case.

All in all, the paper presents interesting results but the reported experimental settings make it harder than necessary to understand the real impact of CP.

----
Score raised +1 based on the promise to improve baselines in the revised version of the paper.


**Time Spent Reviewing:**

5

---

> ### Author Response · Authors · 2021-08-10
> **Clarifications**
>
> Thank you for reviewing our work. We appreciate your time and comments. We understand your concerns on our experimental setting and we hope the following clarification/discussions would help  ease them. First, on the concern about the number of trainable parameters. For the first two easier tasks, CP with $n_p=2$ is only applied to one layer in each model, which only doubles the number of weights in that single layer. In our preliminary parameter tuning runs for the baselines, we made sure that the size of the original baseline is large enough for the task such that even doubling it in a naive manner will not improve the accuracy noticeably. A parameter study for the baselines can be included into the appendix to address this concern. For the third task, we have designed the models to have a similar number of parameters (e.g. 1.28M for CP-GNet5, 1.22M for GNet10). This number will be mentioned in the paper.
>
> Your suggestions on the externally reported baselines are very helpful. In the Burgers setting, we agree that it would be useful  to compare a formal mathematical model with a machine learned set of coefficients as in the suggested reference, a blackbox MLP, and our CP modification that makes a better utilization of known numerical terms than a pure blackbox. We have performed more tests on exact MeshGraphNets models. In general, we find that it is under-performs our baseline model. We believe that this is because of the following differences between our approach and MeshGraphNets: 1) the computation of source term in reaction is very challenging for pure message-passing type processors, but can be taken care of by our "source term section" (shown in Fig 1) of the processor. 2) The original MeshGraphNets updates a large set of edge features through processor blocks, which is hard to train in our setting of limited training data. We be happy to report the results, while emphasizing that the performance difference could be highly dependent on the problem and the training resources.
>
> We agree (and emphasize) that the rocket combustor problem is perhaps the most complex one in existence in terms of developing surrogates for PDE models, but also hope that this complexity is not held against us. We are happy to add the classic (and less challenging cylinder flow problem), where the CP-GNet is able to predict stably and accurately for more than 1500 steps. But the fact that our framework works well on that problem may not add value (especially given space restrictions). We would also like to note that this paper presents a variety of tasks relevant to computational science (beyond the development of surrogates).
>
> We also appreciate the comments on the writeup, including adding a statement of contributions, and relating more clearly the theoretical advantage of CP and the improvements observed in the experiments. The "numerical discretization" and "physical intuition" are reflected in our choices of parameters and architectures. They include the local size and edge directions for the discretized cells, the diffusion term which is closely related to the closure, the essential high-order source terms in chemical reaction computation, etc.

---

> > ### Comment · Reviewer_o2jT · 2021-08-30
> > **Response to authors**
> >
> > Thank you for the response and the clarifications you provided. The combusting flow problem is an interesting domain to show the practical impact of your method. My concerns are not about the complexity of this setting per se, but rather about the relatively large jump in complexity between that case and the two much simpler ones. Simpler settings are good for building intuition, ablation studies, and direct comparisons against prior work. If you could provide the improved baselines in a revised version of the paper, and add comparisons involving some of the MeshGraphNet cases in the appendix, I think that would be a meaningful improvement which would alleviate some of these concerns, and I'm happy to raise my score accordingly.

---

> > > ### Author Response · Authors · 2021-09-15
> > > **Thank you! Updates on comparison with MeshGraphNets and flow over cylinder**
> > >
> > > We would like to thank you for raising the score. We would like to followup and update you with the comparison we performed against the MeshGraphNets. The comparison is conducted with two tests. The first test is the reacting flow case in Sec 4.3, where we implemented the MeshGraphNets in our pipeline. Node labels distinguishing fluid and different types of boundary cells are added to the node features as required by the MeshGraphNets. Due to the complex physics, we also tested a 256-unit MeshGraphNet model, which is twice the width of the original 128-unit model. The 1-/50-/400-step rollout RMSE are {$0.29, 6.8, 46.1$}$\times 10^{-3}$ for our model, {$0.41, 10.8, 58.4$}$\times 10^{-3}$ for the 256-unit MeshGraphNets, and even higher for the original 128-unit model. Visualization results are also  better for our model.
> > >
> > > The second test is the incompressible flow over a cylinder case from the MeshGraphNets work. We implemented our model with their official open-sourced code set and data. The 1-/50-/600-step rollout RMSE for the velocity components are {$3.3, 12.4, 62.5$}$\times 10^{-3}$ for our model, and {$2.1, 8.7, 68.5$}$\times 10^{-3}$ for the MeshGraphNets. From visualizations for randomly selected trajectories, we observed that our model shows clear advantages in the unsteady ones (with oscillating downstream flow), whereas the MeshGraphNets perform slightly better in the steady ones (with a straight downstream flow).

---

### Official Review · Reviewer_ifyj · 2021-07-18

**Rating:** 5
**Confidence:** 4

**Summary:**

This paper proposes methods for neural simulation of physical systems on graphs. In particular it proposes the use of hyper-network style layers (CP-dense) where the weights of the layer are determined by regressing some context parameter. These layers are combined in the CP-GNet architecture, which shares a similar encoder / process / decode structure to MeshGraphNets. The proposed layers are shown to yield performance improvements on simulation tasks relative to dense layers.

**Limitations And Societal Impact:**

The authors have discussed some of the limitations: particularly the parameter count scaling with the dimensionality of context parameters. I think the limitations of the evaluation (mentioned above) haven’t been addressed adequately. I agree with the authors that the potential for negative societal impact of the work is low.

**Main Review:**

The paper’s main contribution is the use of hypernetwork-style layers (CP-dense) in the context of neural simulations on meshes. CP-dense layers are used primarily to integrate heterogeneous input sources. While I don’t believe this is a particularly novel architectural innovation, its effectiveness in this setting is useful to study. It may be of interest to explore other ways to combine inputs, e.g. gating or attention mechanisms. ‘Multiplicative interactions and where to find them’ (Jayakumar et al, 2019) surveys some of these methods. It would also be of interest to evaluate the architecture on more standard graph processing tasks to see if similar gains can be achieved.

The tasks look reasonable, and in particular predicting reacting flows may be of interest to the community.

For the first two tasks, the baseline models with dense layers have fewer parameters than the CP-dense versions, due to the additional parameters introduced in the CP layers. This is problematic given the very small models used, as additional parameters are likely to have a large impact on function approximation capacity. For the DDP models in the first task, the baseline has access to less information than the proposed model. I’d expect the appropriate baseline to concatenate the u and q values in the inputs. For these reasons it’s hard to determine the additional expressive capacity granted by CP layers beyond an addition of model parameters / information.

For the reacting flows task, the baseline method is a variant of the proposed CP-GNet architecture that replaces the CP layers with standard dense layers. This baseline is described as being similar to MeshGraphNets, and uses larger hidden sizes / a greater number of layers, presumably to make the overall parameter counts similar. It would be helpful to report the overall number of parameters for the models used, as it’s hard work to determine if the counts are actually similar.

The evaluation would be stronger if the proposed model was compared prior work such as MeshGraphNets itself and other baselines on a common task. Otherwise we can’t really determine which model is more performant in general.


Overall, I think the proposed CP-dense layer and the associated architecture CP-GNet are useful contributions to the mesh-based simulation field; more work needs to be done to evaluate the methods against parameter-normalized baselines, or against alternative work on common tasks.

Minor critique: I found some parts of the introduction a bit vague, in particular the lines:

[l28-31] MLPs and autoencoders typically observe the solution over the computational domain as a single entity (for every time instance), and the physical quantities from different parts are processed as entangled vectors through hidden layers

[l34-35] Current models often ignore the interactions or hierarchical relations between input features and process them as concatenated mixtures

I now understand that the authors are contrasting hypernetwork-style mixing of inputs from heterogeneous domains, with simple concatenation methods. But on first read I didn’t get it, so I think this paragraph could do with a bit of a re-work.


**Time Spent Reviewing:**

2

---

> ### Author Response · Authors · 2021-08-10
> **Clarifications**
>
> Thank you for reviewing our work. We appreciate your time and comments, which provide valuable directions for future work. We would like to bring up a few more clarifications/discussions below.
>
> Your points on providing a more strict comparison, controlling the network inputs, the number of parameters and with better baselines are well taken. We left out q from the DDP model  to retain the choice of the original implementation. A test of a parallel version of DDP with concatenated u and q as the input was  conducted before the submission, where very limited improvement over the standard version was observed. We are more than happy to add it in the appendix.
>
> We have also performed more tests on exact MeshGraphNets models and we find that - in general - it under-performs our baseline model. We believe that this is because of the following differences between our approach and MeshGraphNets: 1) the computation of source term in reaction is very challenging for pure message-passing type processors, but can be taken care of by our "source term section" (shown in Fig 1) of the processor. 2) The original MeshGraphNets updates a large set of edge features through processor blocks, which is hard to train in our setting of limited training data. We would be happy to report the results in the appendix, emphasizing that the performance difference could be highly dependent on the problem and the training resources.
>
> On the concerns for network sizes, for the first two  tasks, CP with $n_p=2$ is applied to only one layer in each model, which only doubles the number of weights in that single layer. In our preliminary parameter tuning runs for the baselines, we made sure that the size of the original baseline is large enough for the task such that even doubling it in a naive manner will not improve the accuracy noticeably. A parameter study for the baselines can be included into the appendix to address this concern. For the third task, we have designed the models to have a similar number of parameters (e.g. 1.28M for CP-GNet5, 1.22M for GNet10). These exact numbers will be updated.
>
> Admittedly, exploring other mechanisms as suggested, as well as more standard graph processing tasks would be interesting and help to expand the scope of the work. However, we would like to stress that the idea of CP is not limited to just serving specific components as gating or attention. Indeed, we are exploring the possibility of applying CP on top of attention mechanisms in a separate work, which takes global encodings as the parameters. The application of our method is also not limited to graphs. Although in terms of results, the direct simulation on a graph is the highlight of our work, but closures and super-resolution also compose important parts of scientific computing. Limited by the length of the paper, it is hard to further expand discussions on the other, although attractive, aspects.
>
> Your comments on the writing for the introduction are also appreciated.

---

> ### Author Response · Authors · 2021-09-15
> **Updates on comparison with MeshGraphNets and flow over cylinder**
>
> We would like to followup and update you with the comparison we performed against the MeshGraphNets in response to your comments. The comparison is conducted with two tests. The first test is the reacting flow case in Sec 4.3, where we implemented the MeshGraphNets in our pipeline. Node labels distinguishing fluid and different types of boundary cells are added to the node features as required by the MeshGraphNets. Due to the complex physics, we also tested a 256-unit MeshGraphNet model, which is twice the width of the original 128-unit model. The 1-/50-/400-step rollout RMSE are {$0.29, 6.8, 46.1$}$\times 10^{-3}$ for our model (2.5M parameters), {$0.41, 10.8, 58.4$}$\times 10^{-3}$ for the 256-unit MeshGraphNets (7.1M parameters), and even higher for the original 128-unit model (1.8M parameters). Visualization results are also better for our model.
>
> The second test is the incompressible flow over a cylinder case from the MeshGraphNets work. We implemented our model with their official open-sourced code set and data. The 1-/50-/600-step rollout RMSE for the velocity components are {$3.3, 12.4, 62.5$}$\times 10^{-3}$ for our model, and {$2.1, 8.7, 68.5$}$\times 10^{-3}$ for the MeshGraphNets. From visualizations for randomly selected trajectories, we observed that our model shows clear advantages in the unsteady ones (with oscillating downstream flow), whereas the MeshGraphNets perform slightly better in the steady ones (with a straight downstream flow).

---

### Decision · Program_Chairs · 2021-09-27

**Decision:**

Accept (Poster)

**Comment:**

3 of 4 ratings were "accept". A key concern shared by several reviewers was about comprehensive experimental baselines, and the authors responded by performing requested experiments which supported their results. Because the one reviewer who gave the lowest rating (a 5) suggested this experiment, the authors performed it and it was favorable, and the reviewer didn't respond, I'm inclined to treat the reviewer's rating as a 6.